# The Key Technologies of Road Elevation Detection Based on Sensor Fusion

**DOI:** 10.3390/s22155756

**Published:** 2022-08-01

**Authors:** Jin Han, Jia Liu, Hongmei Chang

**Affiliations:** School of Mechanical Engineering, Shaanxi University of Technology, Hanzhong 723001, China; liujia@snut.edu.cn (J.L.); chm130002@snut.edu.cn (H.C.)

**Keywords:** road undulation elevation, data fusion detection, GPS RTK detection, Kalman filtering

## Abstract

The detection of long-distance pavement elevation undulation is the main data basis for pavement slope detection and flatness detection, and is also the data source for 3D modeling and quality evaluation of pavement surfaces. The traditional detection method is to use a level and manual coordination to measure; however, the detection accuracy is low and the detection speed is slow. In this paper, the high-speed non-contact vehicle-mounted road undulationelevation detection method is adopted, combined with the advantages of each sensor measurement; three methods are proposed to detect the road undulation elevation: rotary encoders, accelerometers, attitude sensor data fusion detection; GPS RTK detection; and Kalman filtering detection. Through modeling and experimental comparison, Kalman filter detection is not disturbed by the environment, and the detection accuracy is higher than the current international standard.

## 1. Introduction

With the continuous development and expansion of highways, the detection of road surface diseases has increasingly become an important basis for maintaining and ensuring smooth and safe traffic. Pavement elevation detection is the main indicator for evaluating road surface smoothness. Common detection methods used to measure the elevation of road surfaces involve manual leveling, including the pressure elevation measurement method, the triangular elevation measurement method, and the leveling measurement method [1]. Among them, the leveling method is the artificial elevation measurement method with the highest accuracy at present [2]. The method of leveling the undulating elevation of the road is to manually use the horizontal sight provided by the level to read the level perpendicular to any two points on the measured road section [3]. From the reading on the ruler, the height difference between the measured points is calculated to obtain the elevation. This method is widely used in the elevation measurement of airport runways, highway pavements, and high-speed pavements, as well as pavement construction measurement and engineering surveys.

Since the level adopts the manual segmented measurement method, the following problems are prone to occur:(1)In the measurement process, manual participation is required throughout the process, so the measurement section must be fully closed during the measurement, which affects the normal use of the road, and the measurement efficiency is low;(2)It is necessary to carry a large number of scales and levels when measuring complex road surfaces with long distances, many curves, and many ramps. In the process of measurement, there will be various uncertain errors due to segment distance, ruler placement position, instrument error, manual reading, etc., which cannot guarantee the accurate measurement of the undulating elevation data of the road surface [4]. The measurement accuracy can only reach the centimeter level;(3)It has a high standard for the measurement environment, and it is impossible to measure the road surface in a bad environment such as night, rain and snow, wind and sand, or fog and haze [5].

In recent years, with the continuous development of automated inspection technology, various road comprehensive inspection vehicles have been gradually applied in road inspection operations [6], such as the Japanese Ko-matsu system, the American DHDV system, the Chinese CiCs detection vehicle [7], and so on. Compared with manual detection, the detection efficiency of road comprehensive detection vehicles is significantly improved. At the same time, because road comprehensive inspection vehicles integrate automatic inspection equipment (such as industrial area scan cameras, laser radar, and other non-destructive inspection sensors), the accuracy and reliability of the inspection results are better than manual inspection, but due to the image and laser radar and other equipment, the collected information is easily disturbed by the environment, such as light, weather, etc., and the error is large. This type of detection vehicle is mainly used to detect road diseases such as potholes, cracks, foreign objects, etc., and cannot accurately detect a road’s undulating elevation.

Because of the problems existing in the current methods of road undulation elevation detection, the manual measurement of the leveling instrument can only be used as a benchmark reference during measurement, while the existing automatic road undulation elevation detection equipment has low accuracy and cannot fully meet the detection accuracy and speed requirements. Therefore, developing a high-speed and accurate road elevation detection system suitable for different road surfaces is the key problem that needs to be solved in today’s road detection.

This paper adopts the high-speed non-contact vehicle-mounted elevation detection method, based on the inertial navigation detection principles of aircraft [8]. The attitude sensors, photoelectric rotary encoders, accelerometers, GPS, and other equipment are reasonably matched [9,10]. The data fusion method is used to detect a road’s undulating elevation. Complementary filtering and Kalman filtering are used to optimize the collected data to improve the detection accuracy of road elevation undulation. Due to the use of vehicle-mounted non-contact detection, this detection method can ensure the safety of personnel during detection and does not affect normal traffic; it can be detected without environmental influence (rain and fog), and it can solve the problem of low detection efficiency and low detection accuracy of Manual level detection mode.

## 2. Main Research Content

### 2.1. Sensor Fusion Detection Principle

In this paper, a rotary encoder is installed on the wheel of a road detection vehicle to detect the real-time displacement mileage of the vehicle. A vehicle-mounted attitude sensor [11] measures the pitch angle of the vehicle during the process of driving. As for the vehicle on the road being measured, the attitude sensor at a 200 HZ frequency on the acquisition of ups and downs and connection with the rotary encoder are used to obtain the pitch angle in real time [12], based on the displacement range and real-time mathematical modeling of the vehicle’s pitch angle [13], real-time-calculated road elevation, pavement fluctuation waveform curves, and calculation of the longitudinal slope of the road. Its principle is shown in Figure 1.

As can be seen from Figure 1, when the rotary encoder and the vehicle’s measurement system are used to drive on the measured road surface, the displacement mileage *S* and the angle *θ_x_* between the moving direction and the horizontal direction of the vehicle can be obtained in real time, namely, the pitch angle of the vehicle. It can be defined as:(1)$$\dot{h(t)}=V(t)\sin(\theta x)$$

In Equation (1):*V*(*t*): the traveling speed of the vehicle; θx: vehicle pitching angle, the angle between the moving direction of the vehicle and the horizontal plane;*h*_(_*_t_*_)_: the relative elevation at a time *t*.

As you can see from Figure 1:(2)h=h(0)+h(t)

In Equation (2):*h*: absolute elevation; *h*(0): the reference elevation of the test vehicle.

Because:(3)$$V(t)=\dot{S(t)}$$

In Equation (3):*S*(*t*): the detection distance traveled on the road at time *t*; 

Therefore, based on the above conditions, it can be concluded that:(4)$$h=h(0)+h(t)=h(0)+\int_{0}^{t}\dot{S(t)}\sin(\theta x)dt=h(0)+\underset{\Delta t\to 0}{\lim}\sum_{j=0}^{t}\left((s(j+\Delta t\right)-s(j)\sin(\theta x))$$

In the actual road detection process, the setting of sampling frequency has a great influence on the collection of pavement data. In theory, the higher the sampling frequency and the more sampling points, the higher the accuracy of the data. To ensure the consistency of sampling frequency, the pulse count of the photoelectric encoder for measuring distance *S* and the sampling frequency of the attitude sensor for measuring vehicle pitch angle are set at 200 Hz, that is, $\Delta t=\frac{1}{200}=0.005\;s$, to ensure the synchronization of mileage and elevation measurement, thus obtaining the elevation h at a certain time *t*. According to the undulating elevation h of the road surface, on the one hand, the longitudinal road shape curve along the road surface can be obtained; on the other hand, the longitudinal slope of the road surface can also be obtained through calculation. 

According to the national standard, the slope is the ratio of the road surface elevation difference to the horizontal distance.
(5)$$i=\frac{\Delta h}{100l}\times 100\%$$

In Equation (5):*i*: slope;Δ*h*: elevation difference;*l*: horizontal distance of road surface.

### 2.2. Error Analysis and Treatment

In the selection of a photoelectric rotary encoder, the type with the highest sampling rate possible should be selected, so that data collection during detection is relatively intensive. In the detection process, the undulating elevation of the road surface is calculated by the superposition of the vehicle mileage. The detection error of the attitude sensor on the road pitch angle will be generated, and the cumulative error will appear after continuous superposition calculation.
(6)$$\delta hz=10^{2}S\sin(\delta \theta x)$$

In Equation (6):δhz: elevation error of road surface; *S*: distance of the road surface;δθx: pitch angle error of attitude sensor.

According to the analysis of the equation, the main source of error is the pitch angle error of the attitude sensor. At present, the measurement error of the commonly used vehicle attitude sensor is between 0.5° and 1°. When the error of the pitch angle deviates to a certain direction, the longer the distance of the detected road surface is, the more the cumulative error of the pavement surface elevation will increase, along with the increase of the number of stacking. Therefore, to ensure the accuracy of detection, the error of the attitude sensor should be reduced.

The complementary filtering algorithm [14,15] can eliminate the accumulated error of the attitude sensor. When measuring the pitch angle of the road surface, a low-frequency accelerometer is added to detect the pitch angle of the vehicle when the vehicle is running [16,17], and the pitch angle calculated by the accelerometer is used to correct the measurement value of the attitude sensor. The high-frequency advantages of the attitude sensor and the low-frequency advantages of the accelerometer are effectively combined to reduce the measurement error when measuring alone. The principle is shown in Figure 2.

The specific steps are as follows:

**Step 1**: The attitude sensor obtains the pitch angle θxz according to the integral of the detected angular velocity *ω_x_* against time *t*.
(7)$$\theta xz=\int_{0}^{t}\omega x(t)dt$$

**Step 2**: First, filter out the high-frequency noise of acceleration with low-pass filtering, and then solve the pitch angle θxa by using the accelerometer in the horizontal and vertical directions.
(8)$$\theta xa=\arctan\frac{ax}{az}\times \frac{180}{\pi }$$

**Step 3**: Calculate the difference between the two measured values θxe, correct the attitude sensor by the product of it and the angle compensation coefficient *K*, and finally, obtain the accurate pitch angle of the test vehicle θx minus
(9)θx=θx+k×θxe

In Formula:(10)θxe=θxz−θxa

*k:* compensation coefficient, its value is estimated according to the acquisition frequency of the acceleration sensor, and the value of *k* can be adjusted according to the filtering effect during the actual detection.

### 2.3. Example Verification

To verify the accuracy of the above complementary filtering measurement method, two different roads were selected, the level was manually measured, and the uniform speed of the detection vehicle was automatically measured to complete the detection of elevation and slope.
(1)Elevation measurement

Taking a road in Jining, Shandong Province as the detection object, two methods of artificial level measurement and sensor measurement are adopted to detect the road elevation. As can be seen from the data in Table 1, the detection results of the two methods are very close, and the error between the two methods is less than 4%.
(2)Slope measurement

The longitudinal slope of the runway of an aircraft flight test field in Shaanxi was detected. The total length of the runway was about 1 km, and the designed longitudinal slope was no more than 0.2%. The slope of 11 piles of the runway was detected by artificial level measurement and sensor measurement. The longitudinal slope curve of the runway is shown in Figure 3, and the data are shown in Table 2.

It can be seen from Figure 3 and Table 2 that the road surface fluctuation curve generated by the sensor measured after the complementary filtering is very close to that generated by the data measured by the level [18], both of which reflect the road surface fluctuation. However, when the sensor detects the road surface, the data collection frequency is high, the detection speed is faster, and the detection accuracy error is less than 1 cm. Therefore, the attitude sensor is more accurate and efficient in detecting the undulating elevation of the road surface.

## 3. GPS System for Road Undulation Elevation Detection

The vehicle’s GPS (Global Positioning System) can not only locate and detect the longitude and latitude coordinates of the vehicle, but can also detect the real-time elevation of the road surface [19]. However, conventional GPS can only detect statically or obtain the elevation data of the surface through calculation after the detection, which is relatively inefficient. In this paper, RTK (Real Time Kinematic) positioning technology [20,21] is added to enable the system to realize dynamic detection, real-time completion of the collection of road undulation elevation data, and improve the accuracy and detection efficiency of road detection.

### 3.1. GPS RTK Detection Principle

The positioning principle of RTK is to use carrier-phase technology [22] to process the carrier-phase observation values of two monitoring stations in real time to complete the dynamic positioning [23]. In this paper, one GPS receiver is set as the reference station, and another GPS receiver is installed on the detection vehicle. A base station and detection vehicle are used at the same time to receive the same satellite launch data; the base station will be the benchmark of the location information and the known location information, calculate the error, collect the results through a wireless data link to the detection vehicle, and correct the detection vehicle’s real-time location information, calculating the real-time road surface elevation data [24,25]. The undulating elevation data of the road surface are obtained by the fitting method. The principle is shown in Figure 4.

### 3.2. Case Verification

Through several actual road surface detection experiments, it can be found that the detection accuracy is relatively high under the condition that the reference station is positioned accurately, the reference station can maintain wireless communication with the detection vehicle in real time during the detection process, and the GPS receiver of the reference station and the detection vehicle locks the same regional communication satellite. In the case of an open road section and a good GPS satellite signal, by comparison, the GPS installed by the detection vehicle can be used to calculate the undulating elevation of the road surface, and the measurement results of the leveling instrument are very close to each other [26]. The test results are shown in Figure 5.

When the satellite signal is poor or there are tunnels, electromagnetic interference, shady roads, and some extreme weather, the road fluctuation data detected by the GPS and the actual numerical error are large: causing the distortion of data, as can be seen in Figure 6.

When the GPS signal is good, the data is reliable, and when the GPS signal is poor, the data is distorted.

### 3.3. Error Analysis

Through many tests, it is found that when the satellite signals used for GPS analytical operation are strong and the number of satellites involved in the operation is large (≥5), the satellite information returned by the GPS has higher accuracy, and the measured road elevation fluctuation data is reliable with an error of ±5 cm. However, if the number of satellites involved in the solution is small or the GPS signal cannot be received, the GPS cannot be used to measure the undulating elevation of the road surface. Therefore, for roads with poor GPS signals, such as tunnels, shady roads, mountain roads, etc., GPS detection cannot be carried out, and GPS detection of road fluctuation is limited.

## 4. Kalman Filter Error Optimization

From the perspective of sensor and GPS detection methods and data accuracy, both methods can meet the requirements of high-speed vehicle-mounted road elevation detection, but the two methods also have their advantages and disadvantages, as follows: 

(1) Posture sensor fusion and rotary encoder data sampling frequency are high, the continuity of the data is good, low dependence on the environment, and using the complementary filter reduces the detection of cumulative error, but still, pitch angle theta x has a certain error; for a long, long time of pavement testing and data processing, this error will be accumulated;

(2) GPS uses RTK technology, so the error is small, the data processing is fast, the accuracy is high, and it is not affected by the detection distance and time. However, GPS is highly dependent on the satellite signal and the sampling frequency is low (about 20 Hz), in road sections with poor satellite signal and in the extreme environment, GPS data will produce distortion.

Given this situation, filtering out the noise of the two detection methods and combining the advantages is an optimization scheme to improve the detection accuracy of road undulation elevation [27]. Therefore, it is necessary to use the method of data fusion to combine the data results of the two detection methods to make the detection results more reliable. Compared with common data fusion methods and noise generation rules, this paper adopts the Kalman filter error optimization method [28] to realize the detection of road elevation undulation.

### 4.1. Kalman Filter Technology Route

As shown in Figure 7, the technical route of data fusion is realized in the following steps:

**Step 1**: First detect the GPS signal. When the GPS data is effective, the Kalman filter is used to filter the collected data of the two sensors to obtain the elevation undulation data of the road surface;

**Step 2:** When the GPS data is detected to be invalid, locate the last point where the GPS data is valid. From this point, use complementary filtering and the pitch angle integral algorithm to solve the collected data of attitude sensor and rotary encoder to solve the road undulation elevationdata;

**Step 3:** During the period when GPS data is invalid, the elevation undulation data of the attitude sensor obtained by the pitch angle integral algorithm are connected to the effective starting point of GPS data by the method of linear fitting, and then the Step 1 in the data filtering method is used to calculate the elevation.

### 4.2. Implementation Method

When the data collected by GPS is valid, the Kalman filter is used to optimize the elevation of the road surface by the two detection methods of sensor and GPS [29]. The known quantities obtained by the algorithm through GPS and sensor measurement are *h_k_* and *hz,* and the two detection values are filtered out by the Kalman filter to obtain the optimal elevation of the road surface *h*.

The system model is built according to the Kalman principle, where the state equation is:(11)$$h_{k}=Ah_{k-1}+w_{k}(k\geq 1)$$

In Equation (11):*h_k_*: GPS-measured road undulation elevation; *A*: system parameter;*w_k_*: system noise, which is generated by GPS error, and the covariance is *Q*;*k*: time.

The measurement equation is:(12)$$h_{zk}=Hh_{k}+v_{k}$$

In Equation (12):$h_{zk}$: the elevation measured by sensor data fusion at time *k*; *H*: measurement parameter; $v_{k}$: measurement noise; sensor noise mainly comes from the attitude sensor detection of the pitch angle *θ_x_*; the covariance is *R*.

The prediction equation of the first step can be calculated according to the above two equations:(13)$$\left\{\begin{array}{c} h_{k}=h_{k-1}\\ P_{k}=P_{k-1}+Q \end{array}\right.$$


$h_{k}$: elevation prediction value at time *k*; $P_{k}$: the covariance of*Q:* process noise, whose value is generated by GPS-detected elevation error, can be calculated according to real-time detection elevation. 


The correction equation can be derived as follows:(14)$$\left\{\begin{array}{c} Kg_{k}=P_{k-1}/P_{k-1}+R\\ h_{k}=h_{k-1}+Kg_{k}\left[hZ_{k}-h_{k-1}\right]\\ P_{k}=\left[1-Kg_{k}\right]P_{k-1} \end{array}\right.$$

In Equation (14):*Kg_k_*: Kalman gain coefficient; *R*: the covariance of the measurement noise, which mainly comes from the measurement error *θ_x_* of the attitude sensor’s pitch angle. *R* can be calculated according to the product of the distance measured by the rotary encoder and the pitch angle. 

The current state can be obtained from the prediction equation in the first step of Equation (13), and then these two values are substituted into Equation (14). The three equations in Equation (14) are iterated and *Kg_k_* is corrected, and finally the optimal at time *k* is obtained, which is used to obtain the road undulation elevation *h***.**

### 4.3. Example Verification of Error Optimization

(1)Elevation detection when GPS signal is good

As shown in Figure 8, a 1000-meter-long ordinary road section is selected, and the detection vehicle runs at a constant speed of 40 km/h. Sensor detection and GPS detection occur at a sampling frequency of 200 Hz, and the elevation data collected by GPS and the elevation calculated by the attitude sensor are filtered on the Matlab platform.

Although the elevation detected by GPS and the elevation calculated by the attitude sensor are of high numerical accuracy, their values fluctuate to a certain extent. After the filtering, the fluctuation of the elevation (black line) of the road surface decreases, the curve becomes smoother, and the error further decreases, which is closer to the actual road shape.
(2)Elevation detection when GPS signal changes

The detection vehicle was targeted at a tree-lined road with a large difference in GPS signals. The detected road section is a gently sloped section of a national road in Hanzhong City. The detection results are shown in Figure 9.

The specific implementation steps are as follows: 

**Step 1.** First, detect the GPS signal, and when the GPS signal is good (GPS ≥ 5), use Kalman filtering to optimize the two elevation data calculated by the sensor and the GPS. The red curve in the third picture of 0–420 m, 610–830 m, and 970–100 m in Figure 9;

**Step 2.** When detecting that the GPS signal is invalid (GPS < 5), record the startand end points of the invalid signal, and use the complementary filtering and pitch angle integration algorithm to calculate the road undulation elevation obtained by the sensor data collected on this road. As shown in the figure, a–b and c–d are two blue curves;

**Step 3.** When the GPS signal is restored to be valid, continue to use Step 1 for calculation.

The black curves in the first and third graphs in Figure 9 are the road undulation elevation curves measured manually with a level, and the blue curves in the first graph are the road elevation undulations curves measured by GPS. When the GPS signal is invalid, the detected values are distorted; the third graph is the blue–red mixed curve for the final optimized road undulation elevation curve. It can be seen that the two curves are very close, indicating that the optimization results are accurate.

### 4.4. Detection Accuracy Analysis

For the detected road surface shown in Figure 9, four methods, including artificial level (detection benchmark), GPS RTK, sensor fusion, and Kalman filtering, were adopted in this paper to detect the undulating elevation of road surfaces. The detection results are shown in Table 3. It can be seen from Table 3 that the maximum error of sensor fusion measurement is 3.9% compared with that of artificial level measurement, although the pitch angle detection error of the attitude meter sensor after complementary filtering is ±0.2°. However, with the increase of the detection distance, the error will accumulate, and the error is related to the distance of the road surface measured, namely:(15)$$\delta hz=10^{2}S\sin(\delta \theta x)=10^{2}S\sin(0.2^{{}^{\circ}})$$

It is assumed that the pitch Angle error of the attitude sensor is 0.2° in the same direction. According to Equation (15), when the detection distance *S*(m) is longer, the error tends to increase gradually. Therefore, this method can only be used for medium- and short-distance road detection.

**Table 3 sensors-22-05756-t003:** Comparison of detection data.

Node	Distance (m)	Level	Sensor Fusion		GPS RTK Measured		Kalman Filter Data Fusion	
		*h* (cm)	*h* (cm)	Error (%)	*h* (cm)	Error (%)	*h* (cm)	Error (%)
0	0	25	24.5	2.0	25	0	25.2	0.8
1	100	32	33.1	3.4	33	3.2	32.6	1.8
2	200	43	44.7	3.9	45	4.6	43.5	1.1
3	300	51	52.1	2.1	52	1.9	51.8	1.5
4	400	72	73.9	2.6	74	2.7	73.4	1.9
5	500	84	85.2	1.4	92 (distortion)	9.9	85.5	0.4
6	600	91	93.3	2.5	101 (distortion)	10.9	92.5	1.6
7	700	100	98.2	1.8	102	2	101.3	1.3
8	800	112	109.5	2.2	114	1.7	113.4	1.3
9	900	122	125.5	2.8	113 (distortion)	7.3	123.7	1.4
10	1000	131	134.6	2.7	135	3.0	134.3	1.8

When using GPS RTK detection and GPS signal is good, the road elevation maximum error level is 4.6% compared with that of artificial level measurement. When the GPS signal is poor (the number of satellites < 5), the maximum error is 10.9%, and the data is distorted. In theory, although the maximum error of a GPS signal is ±5 cm when it is good, it cannot be detected for some blocks and signal interference due to over-reliance on signal quality.

Compared with the artificial level, the maximum error of the Kalman filter is 1.9%, and the average error is 1.35%, which is less than that of sensor fusion. The accuracy level can reach millimeter level (the artificial level is centimeter-level), and the detection accuracy will not be reduced because of the increase in detection distance and the fluctuation of the GPS signal. Therefore, the Kalman filter method is suitable for long-distance road surface elevation undulation detection in various environments, and the detection accuracy and reliability are high.

## 5. Conclusions

This paper proposes a sensor data fusion detection method for the detection of road elevation undulation. The specific research contents are as follows: 

(1) The advantages and disadvantages of the measuring method of road undulation elevation are discussed. The detection method of road undulation elevation based on sensor data fusion is proposed, and the technical realization route is designed;

(2) The complementary filtering algorithm of sensor data fusion based on road elevation fluctuation is designed to improve the data accuracy of the attitude sensor. The complementary filtering method is designed to eliminate sensor error. The Kalman filter is used to optimize the sensor and GPS data error, to realize the accurate detection of different road elevation fluctuations in a complex environment. The accuracy and reliability of the data fusion detection method are verified by an example; 

(3) The data error and precision of the detection method in this paper are analyzed and compared, and the accuracy and reliability of sensor fusion detection of road undulation elevation are verified;

(4) In this paper, a comparative verification method is used for the verification of the test results. Although this method can ensure the reliability of the results, the verification process takes a lot of time, is inefficient, and is prone to errors. The authors will add functionally validated machine learning methods [30] and observational modeling algorithms [31] in future research to improve the efficiency and accuracy of validation.

## Figures and Tables

**Figure 1 sensors-22-05756-f001:**
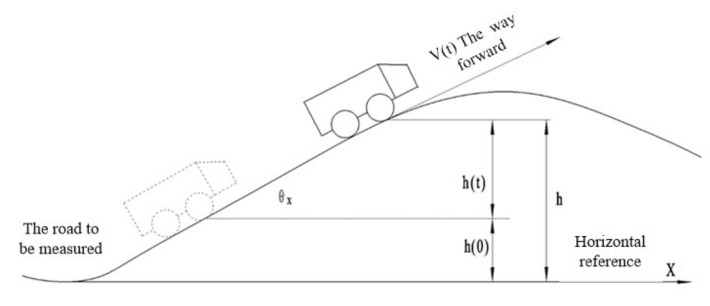
Working principle of road undulation elevation measurement.

**Figure 2 sensors-22-05756-f002:**
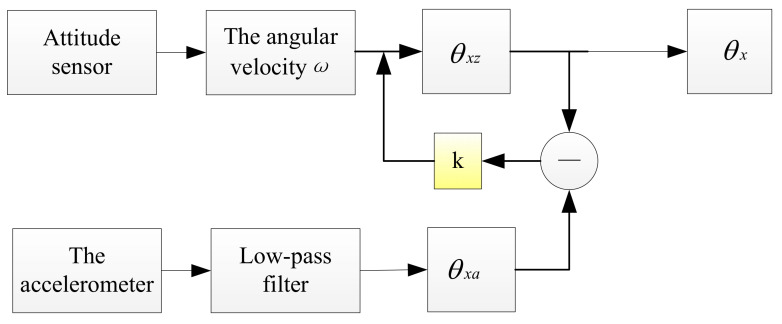
Schematic diagram of pitching complementary filtering.

**Figure 3 sensors-22-05756-f003:**
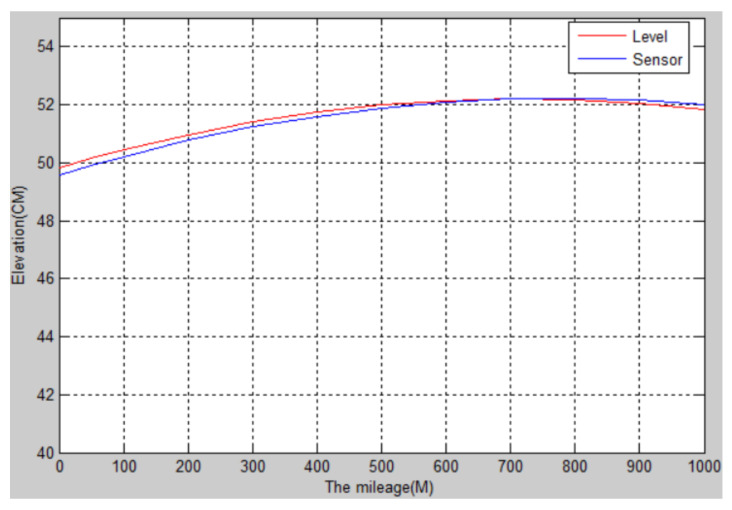
The longitudinal slope of the runway of an aircraft test flight ground.

**Figure 4 sensors-22-05756-f004:**
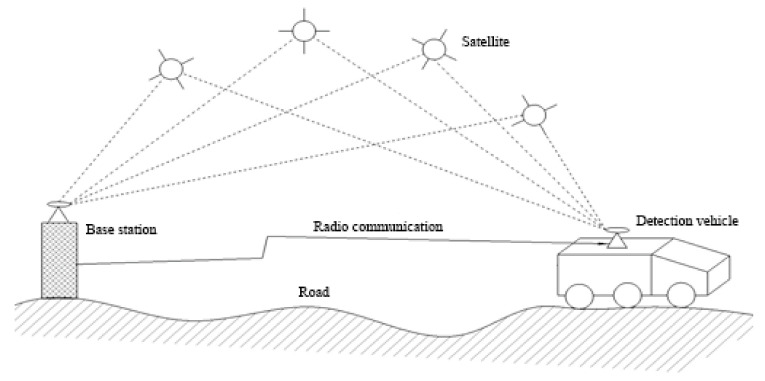
Principle of RTK detection.

**Figure 5 sensors-22-05756-f005:**
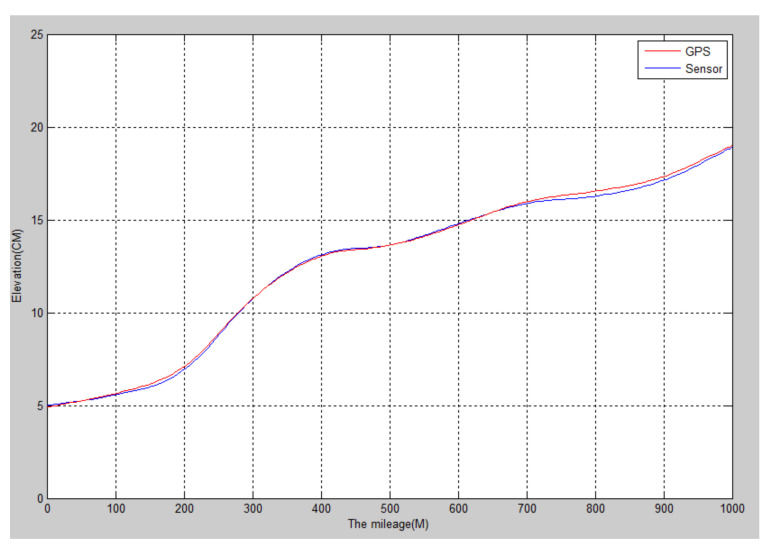
GPS and level road undulation elevation detection curve.

**Figure 6 sensors-22-05756-f006:**
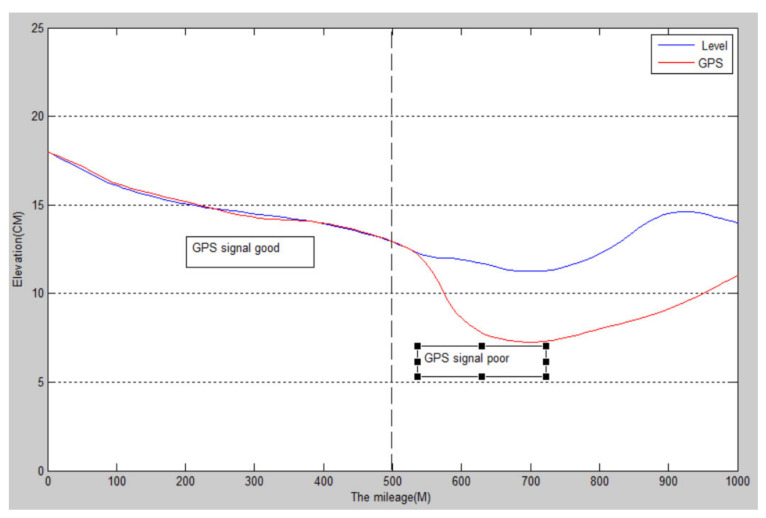
Comparison of measured data of road undulation elevation.

**Figure 7 sensors-22-05756-f007:**
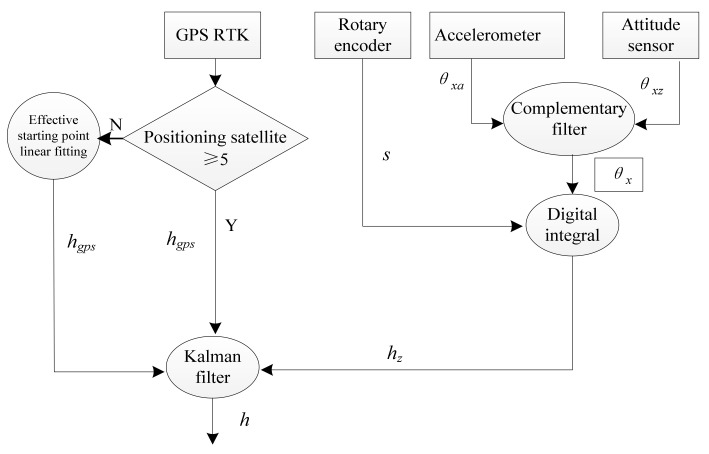
Technology route.

**Figure 8 sensors-22-05756-f008:**
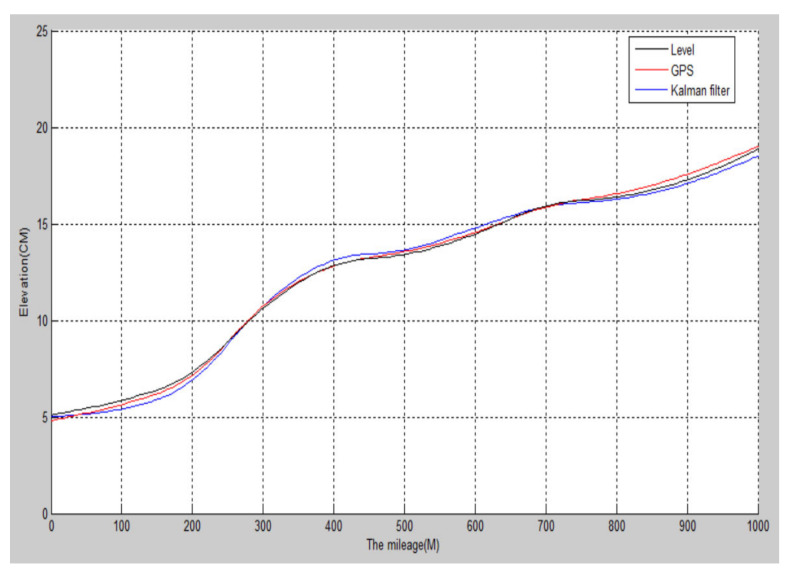
Kalman filter elevation contrast.

**Figure 9 sensors-22-05756-f009:**
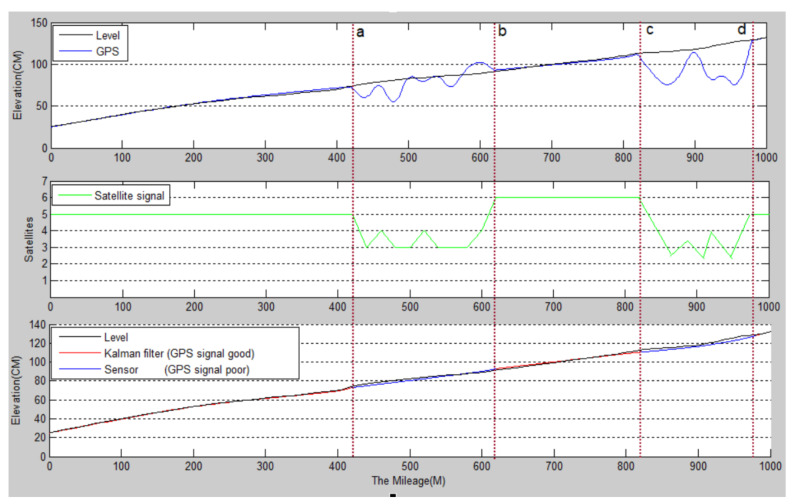
The undulating waveform of road surface obtained by the data fusion algorithm.

**Table 1 sensors-22-05756-t001:** Elevation data of a road in Jining, Shandong Province.

Measure the Node	Distance (m)	Level Elevation (cm)	Attitude Sensor Elevation (cm)	Error (%)
TP1	50	26	25.2	3.0
TP2	50	56	57.3	2.3
TP3	50	86	87.5	1.7
TP4	50	105	102.5	2.3
TP5	50	147	151.6	3.1
TP6	50	149	154.4	3.6
TP7	50	152	156.5	2.9
TP8	50	115	110.8	3.6
TP9	50	102	105.5	3.4
TP10	50	105	107.8	2.8

**Table 2 sensors-22-05756-t002:** The slope of the runway on test flight ground.

Measure the Node	Distance (m)	Level		Sensor		Slope Error (%)
		Δ*h* (cm)	Slope (%)	Δ*h* (cm)	Slope (%)	
0–1	100	1	0.01	3	0.03	0.02
1–2	100	2	0.05	4	0.04	0.01
2–3	100	1	0.01	0.5	0.005	0.0015
3–4	100	7	0.07	6.7	0.067	0.003
4–5	100	1	0.01	2.8	0.028	0.018
5–6	100	4	0.04	5	0.05	0.01
6–7	100	5	0.05	3	0.03	0.02
7–8	100	10	0.10	9.3	0.093	0.007
8–9	100	2	0.02	4.7	0.047	0.0027
9–10	100	1	0.01	0.2	0.002	0.008

## Data Availability

Not applicable.

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
