# Peer review of "The Key Technologies of Road Elevation Detection Based on Sensor Fusion"

_sensors, 2022, doi:10.3390/s22155756_

Round 1

Reviewer 1 Report

The paper is overall exciting and fits within the scope of this journal. The topic (Research on key technologies of road elevation detection based on sensor fusion) is impressive. The focused point of the research is understandable and acceptable, and the proposed technique/algorithm is adequately validated with some results which prove the research's objectives.

The abstract is acceptable, but it can be improved. Although the abstract includes all necessary information (background, purpose, method, results, and conclusion), the results and conclusion are not clear.

Introduction can be improved. Authors have to address how their work is different from existing researchers. The motivation and problem statement of this paper is not clear. The contribution of the paper is not written explicitly. A short literature review and section organization may be added in this introduction.

The presentation of the paper needs to be improved. Although the authors presented 3 main sections, the overall quality of the presentation of this paper can be improved. In section 2, the authors combined many points (the proposed idea, method, results, analysis, and discussions). Authors should separate them into different sections.  

According to the title of the research, road elevation detection based on sensor fusion is considered. To improve the technical concepts, sensor fusion must be dominating the research for keeping the correlation of the paper. So, the authors should add a line about sensor fusion in the abstract.

Although GPS is a popular abbreviation, the author should expand it or define it when authors use it for the first time. Authors should check other abbreviations as well. GPS and Gps are used. Authors should either of these not both.

Although this paper includes relevant technical concepts and the significance of content with necessary results, technical information in some paragraphs is not clear. Example: Section 2.1.2 is not clear.

The conclusions need some improvement. Authors should provide a better overview of the implications of their research for a general audience. The authors may add the future work of the research obtained from your results.

References are not enough

Author Response

Dear Reviewers:

Thank you for taking the time to review my manuscript over the holidays, and providing us with this great opportunity to submit a revised version of our manuscript. We appreciate the detailed and constructive comments provided by the reviewers. We have carefully revised the manuscript by incorporating all the suggestions by the review panel.

We hope this revised manuscript has addressed your concerns, and look forward to hearing from you.

Your Sincerely,

Jin Han

In response to your suggestion, I have made the following changes:

  1. The abstract is acceptable, but it can be improved. Although the abstract includes all necessary information (background, purpose, method, results, and conclusion), the results and conclusion are not clear.

Response1:The Abstract was supplemented and revised to supplement the results and conclusions.

  1. Introduction can be improved. Authors have to address how their work is different from existing researchers. The motivation and problem statement of this paper is not clear. The contribution of the paper is not written explicitly. A short literature review and section organization may be added in this introduction.

Response2:The research purpose and contribution of this paper are added in the introduction section, and the literature review is supplemented.

  1. The presentation of the paper needs to be improved. Although the authors presented 3 main sections, the overall quality of the presentation of this paper can be improved. In section 2, the authors combined many points (the proposed idea, method, results, analysis, and discussions). Authors should separate them into different sections.

Response3: The section 2 of the article has been reorganized and divided into three parts for exposition.

  1. According to the title of the research, road elevation detection based on sensor fusion is considered. To improve the technical concepts, sensor fusion must be dominating the research for keeping the correlation of the paper. So, the authors should add a line about sensor fusion in the abstract.

Response4: Added clarification to the abstract.

  1. 5. Although GPS is a popular abbreviation, the author should expand it or define it when authors use it for the first time. Authors should check other abbreviations as well. GPS and Gps are used. Authors should either of these not both.

Response5: The definition of GPS is explained and the unified format is GPS

  1. Although this paper includes relevant technical concepts and the significance of content with necessary results, technical information in some paragraphs is not clear. Example: Section 2.1.2 is not clear.

Response6: Added relevant technical information.

7.References are not enough.

Response7: Added relevant references.

Reviewer 2 Report

This is a well-written and carefully structured paper that presents a comprehensive characterization of the new technology used for quality evaluation of pavement surfaces. The paper is based on a very strong and detailed analisys of the material studied.

I have a few comments that might be usefully addressed to improve the overall quality of the paper:

-       The term “undulation” is not a proper term for this field, you should use “unevenness” instead.

-       The introduction part should be more detailed with some actual works in the field.

-       The main content of the paper is very well written, but the conclusions need improvement, now they are vaguely written text in order to highlight the results obtained.

-       The experimental program is detailed and the quality of the results’ discussion in its current form satisfies the requirements necessary for a research paper.

The paper is therefore very well suited to this journal. The authors are to be commended on the professional quality of the research and the paper. Also, some thoughts are due on future developments, especially on the practical applicability of this method.

Author Response

Dear Reviewers:

Thank you for taking the time to review my manuscript over the holidays, and providing us with this great opportunity to submit a revised version of our manuscript. We appreciate the detailed and constructive comments provided by the reviewers. We have carefully revised the manuscript by incorporating all the suggestions by the review panel.

We hope this revised manuscript has addressed your concerns, and look forward to hearing from you.

Your Sincerely,

Jin Han

In response to your suggestion, I have made the following changes:

  1. The term “” is not a proper term for this field, you should use “” instead.

Response1: Replace the word " undulation " with " unevenness ".

  1. The main content of the paper is very well written, but the conclusions need improvement, now they are vaguely written text in order to highlight the results obtained.

Response2: The conclusion part has been supplemented and improved

  1. The introduction part should be more detailed with some actual works in the field

Response3: Introductory section supplemented with research findings in this area.

Reviewer 3 Report

The article shows a new methodology to evaluate the pavement surface. This methodology is based on the detection of long-distance pavement undulation elevation, which is the main data basis for pavement slope detection and flatness detection.

Results show that this method can complete the accurate detection of different pavements and meet the detection requirements of pavement quality evaluation.

The article needs major revision before publication. Please pay attention to the following questions.

The written English needs a review in the whole manuscript, please check this.

The introduction section introduces a brief resume of the state of the art of these technologies. This reviewer thinks that this state of the art should be increased.

At the end of section 1, please make a short introduction to the paper structure.

In section 2, please make a short introduction of the section to points 2.1 and point 2.2

Figure 1, please introduce for a better understanding S (distance between initial and final vehicle) and θx (pitch angle of the vehicle or slope).

Authors talk about θx first saying that it is the Angle θx between the moving direction and the horizontal direction of the vehicle. Then say that can be obtained in real time namely the pitch angle of the vehicle. Are the same? Please explain this better.

Please correct the sentence on page 5 of 15 line 179. Please make a better description of the case study, the road, PK, distance analyzed, etc. Table 1 (500 m), table 2 (1000 m). Please rewrite better this part.

Please check equations. Pay attention to equation (13) use brackets and parenthesis.

Figure 8, please use another color for Kalman filter.

Some format errors are detected in author contributions, funding, etc… and reference section, please check again the initial template for the paper.

Author Response

Dear Reviewers:

Thank you for taking the time to review my manuscript over the holidays, and providing us with this great opportunity to submit a revised version of our manuscript. We appreciate the detailed and constructive comments provided by the reviewers. We have carefully revised the manuscript by incorporating all the suggestions by the review panel.

We hope this revised manuscript has addressed your concerns, and look forward to hearing from you.

Your Sincerely,

Jin Han

In response to your suggestion, I have made the following changes:

  1. At the end of section 1, please make a short introduction to the paper structure.

Response1:At the end of the first section of this paper, an introduction to the structure of this paper has been added.

  1. In section 2, please make a short introduction of the section to points 2.1 and point 2.2

Figure 1, please introduce for a better understanding S (distance between initial and final vehicle) and θx (pitch angle of the vehicle or slope).

Response2: In 2.1 and 2.2 of the second section of this article, the introduction of relevant points has been added. The relevant parameters in the formula are introduced in detail, such as S and θx, etc.

  1. Please correct the sentence on page 5 of 15 line 179. Please make a better description of the case study, the road, PK, distance analyzed, etc. Table 1 (500 m), table 2 (1000 m). Please rewrite better this part.

Response3: Corrected sentence on page 15, line 179. Correction and explanation of Equation 13 and parameter Pk have been added. Added relevant descriptions to Table 1 and Table 2.

  1. Figure 8, please use another color for Kalman filter.

Response4: In Figure 8, the applied part of the Kalman filter is marked.

  1. Some format errors are detected in author contributions, funding, etc… and reference section, please check again the initial template for the paper.

Response5: Amendments were made to the relevant formatting proposed by the reviewers.

Reviewer 4 Report

This manuscript is related to a modeling approach. This is an important subject for many real applications in transportation and automotive systems.

The manuscript cannot be published in this form and it must be revised using these comments:

1) A careful editing must be done.

2) You should show in the introduction which are the new ideas of this manuscript with respect to the literature in the field.

3) The motivation of your modeling approach should also be better articulated in the context of the analysis of the literature in the field. This affects the novelty. Your past well-acknowledged papers ought to be inserted in this discussion.

4) In the context of the comment 3), you might consider to include the following topics on classical and new modeling approaches, which proved to be successful in in various fields: Machine learning techniques for improving the performance metrics of functional verification (ROMJIST 2021), Aspects concerning the observation process modelling in the framework of cognition processes (APH 2012), Tensor product-based model transformation approach to tower crane systems modeling (AJC 2021).

5) It is expected to give modeling steps to describe your approach. They should be associated with equations.

6) The pseudocode would be useful as well, also associated with equations and text.

7) How did you solve the optimization problem?

8) The validation and comparison are not clear to me. Additional data is needed. For the comparison to be OK, you are expected to use similar model complexity and the same design specifications and approach. In other words, you are advised to save the programs and datasets in a webpage / repository and cite a link to that webpage in the manuscript body. That will ensure a transparent validation and comparison.

9) The very good results might indicate overfitting.

10) You should better point out the validation of your models.

11) The title is not inspired. You must delete "Research on" and also please be more specific. The current manuscript title leads to the idea of a survey manuscript, which is not the case.

Concluding, the approach has a big potential to be appreciated. These comments will further increase its impact.

Author Response

Dear Reviewers:

Thank you for taking the time to review my manuscript over the holidays, and providing us with this great opportunity to submit a revised version of our manuscript. We appreciate the detailed and constructive comments provided by the reviewers. We have carefully revised the manuscript by incorporating all the suggestions by the review panel.

We hope this revised manuscript has addressed your concerns, and look forward to hearing from you.

Your Sincerely,

Jin Han

In response to your suggestion, I have made the following changes:

  1. You should show in the introduction which are the new ideas of this manuscript with respect to the literature in the field.

Response1: A background description is added in the introduction section, and the purpose and method of this study are proposed

  1. The motivation of your modeling approach should also be better articulated in the context of the analysis of the literature in the field. This affects the novelty. Your past well-acknowledged papers ought to be inserted in this discussion.

Response2: The motivation for this paper has been added in the Introduction section, and other contributions related to this paper are supplemented in Section 2.

  1. In the context of the comment 3), you might consider to include the following topics on classical and new modeling approaches, which proved to be successful in in various fields: Machine learning techniques for improving the performance metrics of functional verification (ROMJIST 2021), Aspects concerning the observation process modelling in the framework of cognition processes (APH 2012), Tensor product-based model transformation approach to tower crane systems modeling (AJC 2021).

Response3: I have carefully read the article you provided and gained a lot. Although the model adopted in this paper still has some flaws, it has been verified in practical application and can solve the problem. Due to time constraints, I am going to use the relevant machine learning algorithm you recommended to apply in future experiments, which will definitely be of great help.

  1. It is expected to give modeling steps to describe your approach. They should be associated with equations.

Response4: The detection steps are supplemented for the experiments involved in Figure 9, and the algorithm for each road segment is explained.

5&7. The pseudocode would be useful as well, also associated with equations and text.The validation and comparison are not clear to me. Additional data is needed. For the comparison to be OK, you are expected to use similar model complexity and the same design specifications and approach. In other words, you are advised to save the programs and datasets in a webpage / repository and cite a link to that webpage in the manuscript body. That will ensure a transparent validation and comparison.

Response5&7: The related software involved in this article has been developed and can be found in (Han, J., Cui, L. & Shi, S. Road rut detection system with embedded multi-channel laser sensor. Int J Adv Manuf Technol .2021.). This article is in Due to the use of national roads and airport runways during the experiment, according to relevant national laws, the data is not allowed to be disclosed, so detailed data cannot be provided, please understand, thank you.

6.How did you solve the optimization problem?

Response6:This paper mainly uses two methods to solve the optimization, one is to use complementary filtering to optimize the acquisition accuracy of the accelerometer, and the other is to use Kalman filtering to solve the road undulating elevation optimization problem.

8&9.The very good results might indicate overfitting. You should better point out the validation of your models.

Response8&9: In this paper, two methods are used to avoid overfitting. One is to increase the number of data collection. In this paper, the acquisition frequency of the sensor is increased as much as possible to obtain more data points under the condition of reasonable matching; the second is to adjust the initial parameters in the filtering algorithm according to the operation results, such as the initial gain coefficient of Kalman filtering, etc., to minimize the error.

10.The title is not inspired. You must delete "Research on" and also please be more specific. The current manuscript title leads to the idea of a survey manuscript, which is not the case.

Response10: Title changed to " The key technologies of road elevation detection based on sensor fusion".

Round 2

Reviewer 1 Report

The revised manuscript has significantly improved, and the comment of previous reviewers had addressed. However, a few modifications need to be addressed to enhance the quality and presentation of the manuscript before the final submission to the journal. Some detailed comments and suggestions can be seen as follows:

1.      Although the introduction is better than before, adding section organization will be improving the presentation and quality of the paper.

2.      Some abbreviations are not defined/expanded. Example: hips (page 10 and line 377). It should be fixed and explained. Authors have to check other abbreviations as well.

3.      The authors corrected the GPS instead Gps in many places but still, the abbreviation “Gps” is used in Figures 5 and 6.  

4.      Some references are added but they have not been highlighted in the revised version of the paper.

Although the revised paper is good and updated with more sections, the above comments will improve the quality of the paper further. It also contains some new contributions, ideas, and novelty with the fusion algorithm/steps and good results. Considering the paper with the above points (as minor corrections) will be improving the overall quality and presentation of the paper.

Author Response

Dear Reviewers:

Thank you for your new suggestions for my manuscript, which I have revised according to your comments, please review. I wish you good health and happiness.

Your Sincerely,

Jin Han

In response to your suggestion, I have made the following changes:

  1. Some abbreviations are not defined/expanded. Example: hips (page 10 and line 377). It should be fixed and explained. Authors have to check other abbreviations as well.

Response1: Corrected the parameter on page 10, line 377.

  1. The authors corrected the GPS instead Gps in many places but still, the abbreviation “Gps” is used in Figures 5 and 6.

Response2: Change the label of 'Gps' to 'GPS' in Figures 5 and 6.

  1. Some references are added but they have not been highlighted in the revised version of the paper.

Response3: Added references are highlighted in red.

Reviewer 3 Report

Thank you for your comments.

The paper is ready for publication. 

Author Response

Dear  Reviewers:

Thank you for your affirmation of my manuscript, and I wish you good health and happiness.

Your Sincerely,

Jin Han

Reviewer 4 Report

The paper is very well revised and deserves to be published.

Author Response

(The authors gave the same response as above.)
